# Endophytic *Trichoderma* Species Isolated from *Persea americana* and *Cinnamomum verum* Roots Reduce Symptoms Caused by *Phytophthora cinnamomi* in Avocado

**DOI:** 10.3390/plants9091220

**Published:** 2020-09-17

**Authors:** Petra Andrade-Hoyos, Hilda Victoria Silva-Rojas, Omar Romero-Arenas

**Affiliations:** 1Manejo Sostenible de Agroecosistemas, Instituto de Ciencias, Benemérita Universidad Autónoma de Puebla, Edificio VAL 1, Km 1,7 Carretera a San Baltazar Tetela, San Pedro Zacachimalpa, Puebla 72960, Mexico; andrad@colpos.mx; 2Producción de Semillas, Colegio de Postgraduados, Campus Montecillo, Km. 36.5 Carretera México-Texcoco, Estado de México 56230, Mexico; hsilva@colpos.mx; 3Centro de Agroecología, Instituto de Ciencias, Benemérita Universidad Autónoma de Puebla, Edificio VAL 1, Km 1,7 Carretera a San Baltazar Tetela, San Pedro Zacachimalpa, Puebla 72960, Mexico

**Keywords:** antagonism, biocontrol, disease incidence, root rot, soil-borne

## Abstract

Avocado root rot caused by the oomycete *Phytophthora cinnamomi* is a severe disease that affects avocado production in Mexico and worldwide. The use of biological control agents such as *Trichoderma* species isolated from places where the disease is always present, represents an efficient alternative to reduce losses. Thus, the objective of this research was to evaluate the biocontrol ability of 10 endophytic *Trichoderma* spp. strains against *P. cinnamomi* tested both in vitro and in the greenhouse. The endophytic *Trichoderma* spp. were recovered from *Persea americana* and *Cinnamomum verum* roots, isolated and purified on potato–dextrose–agar medium. Ten strains were identified by phylogenetic reconstruction of the internal transcribed spacer region of rDNA sequences as *T.*
*asperellum* (T-AS1, T-AS2, T-AS6, and T-AS7), *T. harzianum* (T-H3, T-H4, and T-H5), *T. hamatum* (T-A12), *T. koningiopsis* (T-K8 and T-K11), and *P. cinnamomi* (CPO-PCU). In vitro dual-culture assay, the percentage of inhibition of radial growth (PIRG) between *Trichoderma* spp. and *P. cinnamomi* strains was measured according to the Bell’s scale. PIRG results indicated that T-AS2 reached the highest value of 78.32%, and T-H5 reached the lowest value of 38.66%. In the greenhouse, the infection was evaluated according to the percentage of disease incidence. Plants with the lowest incidence of dead by avocado root rot were those whose seedlings were inoculated with T-AS2 and T-AS7, resulting in only 5% death by root rot caused by *P. cinnamomi*. The disease incidence of seedlings with wilt symptoms and death decreased more than 50% in the presence of *Trichoderma* spp. Relying on the results, we conclude that *T. asperellum* and *T. harzianum* contribute to the biocontrol of soil-borne pathogenic oomycete *P. cinnamomi*.

## 1. Introduction

The avocado (*Persea americana* Mill) is one of the most important fruit worldwide; its demand is increasing in the international market due to its nutraceutical properties and its use as a food supplement [1]. The global production of avocado was approximately 6048 million tons in 2017, of which Mexico produced approximately 583,426 tons on a surface of 188,723 ha^−1^ [2]. Currently, the yields of avocado plantations in the intended areas for this crop are decreasing, mainly due to phytosanitary problems that are difficult to prevent and control [3].

Avocado root rot is the major disease caused by *Phytophthora cinnamomi* Rands; this same oomycete infects a wide host range of wild and cultivated plants worldwide. This disease severely affects avocado plantations representing a threat to natural ecosystems [4,5].

This disease is also known in Mexico as avocado sadness in which the root and feeder roots shown typical root rot symptoms [1,6]. In Mexico, avocado root rot has been reported in the states of Queretaro, Guanajuato, Michoacan, and Puebla, with a mortality that reaches up to 100% attributed to *P. cinnamomi* infection [7].

In conventional avocado production, to mitigate losses caused by this pathogen, the most common practice is the indiscriminate application of high amounts of chemical fungicides and synthetic fertilizers [8], which cause a negative environmental impact on agroecosystems. In contrast, the use of cultural practices, biological control, and resistant rootstocks as a strategy to reduce the incidence of the disease offer sustainable alternatives to *P. cinnamomi* management [9].

The use of filamentous fungi as biocontrol agents represents an effective alternative for agricultural production systems because it reduces the numerous applications of various types of fungicides. For more than eight decades, research has been conducted on the use of different species of *Trichoderma*. Multiple mechanisms of action have been described, but the most relevant are (a) the production of secondary metabolites that strengthen the plant’s immune system [10,11], (b) the action of natural mycoparasites and antagonistic agents [12] and (c) the promotion of seed germination and plant growth, as well as an increased root and foliage biomass and mineral assimilation [13,14,15].

Thus, different indigenous species of *Trichoderma* play an important role in agroforestry and natural ecosystems; they are considered biological control agents because they act against pathogens of agricultural importance, such as *P. cinnamomi* [16]. Investigations of dual in vitro confrontations with different plant pathogens and *Trichoderma* spp. have been frequently conducted [17,18]. However, there are no experiments performed in vitro between *Trichoderma* spp. and *P. cinnamomi* along with inoculated avocado seedlings under greenhouse.

Recently, different species of *Trichoderma* have been evaluated as antagonists of soil-borne, such as *Phytophthora capsici* with *T. harzianum* in pepper [15], *Phytophthora nicotianae* with *T. harzianum* and *T. asperellum* in pepper [19], *Phytophthora melonis* with *T. harzianum* in cucumber [20], *P. cinnamomi* with *T. atroviride* in tomato [18] and the oomycete *Pythium ultimum* with *T. cerinum*. This last antagonist was isolated from avocado roots [12,21]. In this sense, there are several studies related to indigenous species of *Trichoderma* recovered from the plant rhizosphere and their inference in the health of plants. For this study, some indigenous endophytic *Trichoderma* species isolated from avocado and cinnamon roots were tested in vitro against *P. cinnamomi*, and their influence on avocado seedling health under greenhouse conditions.

## 2. Materials and Methods

### 2.1. Sampling of Symptomatic Avocado and Cinnamon Plants

To determine the presence of indigenous species of *Trichoderma* during the spring and summer seasons of 2018, avocado rootstock *P. americana* var. drymifolia roots were recovered from the states of Uruapan, Michoacan (19°23′60.00″ N and 101°57′36.00″ W), Axocopan Atlixco, Puebla (18°54′2.64″ N and 98°27′51.36″ W), Tepatepec Hidalgo (20°14′3.84″ N and 99°5′14.28″ W), and Tepeyanco, Tlaxcala (19°14′42.80″ N and 98°13′44.04″ W), and *Cinnamomum verum* J. Presl roots were recovered from Zozocolco de Hidalgo, Veracruz (20°8′25.80″ N and 97°34′31.80″ W). At each site, roots with rhizosphere soil were collected from cinnamon and avocado plants that showed chlorosis and wilting foliage, five samples were collected at a depth of 0 to 30 cm and kept in plastic bags until further analysis.

### 2.2. Fungal Isolates

The roots from avocado and cinnamon were washed with tap water, followed by immersion in 1% sodium hypochlorite for 30 s, rinsed twice with sterile distilled water, wrapped with sterilized paper towels, and placed in a laminar flow chamber at 20 °C for 15 min to dry. Later, the roots were fragmented into 1 cm pieces and placed vertically by embedding a half centimeter in a Petri plate with potato dextrose agar (PDA, Dioxon) medium amended with chloramphenicol (Sigma-Aldrich, St. Louis, MO, USA) at 20 mg mL^−1^. The plates were maintained at 25 °C for 40 h and then under white light at 20 °C for 3 days to stimulate sporulation above the root not embedded in the PDA medium (Figure 1). After three days, the plates were examined, and they showed colonies with morphological characteristics similar to those of *Trichoderma* spp. They were transferred to fresh PDA to obtain pure cultures by the single conidium technique. A *Phytophthora cinnamomi* (CPO-PCU) was isolated on V8 juice agar medium amended with chloramphenicol at 20 µg mL^−1^ and then incubated in the dark at 28 °C for 72 h. The plates were examined, and they showed morphological characteristics (hyphal swellings, chlamydospore and sporangium) similar to *P. cinnamomi* [5,22].

### 2.3. DNA Extraction, PCR Amplification and Sequencing

This procedure was performed with the 2% cetyl trimethylammonium bromide (CTAB) method according to Doyle and Doyle [23] with some modifications [24]. Genomic DNA was suspended in 100 µL of sterile HPLC water and quantified by spectrophotometry in a NanoDrop 2000 C (Thermo Scientific, Waltham, MA, USA). To determine the DNA quality, absorbance values between 1.8 and 2.2 at A280/260 and A230/260 nm were considered acceptable. Finally, the DNA was diluted to 20 ng µL^−1^ and then stored at −20 °C for PCR amplification.

The internal transcribed spacer (ITS) region of rDNA was amplified for the *Trichoderma* strains with ITS5 (5′-GGAAGTAAAAGTCGTAACAAGG-3′)/ITS4 (5′-TCCTCCGCTTATTGATATGC-3′) and for the *Phytophthora* strain with ITS6 (5′-GAAGGTGAAGTCGTAACAAGG-3′)/ITS4 universal primers [25]. The reaction mixture was prepared in a final volume of 15 µL with 1× *Taq* buffer DNA polymerase, 0.18 µM of each dNTP, 0.18 µL of each primer containing 10 pmol, 0.90 U of GoTaq DNA polymerase (Promega, Madison, WI, USA), and 40 ng µL^−1^ DNA. PCR was performed in a Peltier PTC-200 DNA thermal cycler (Bio-Rad, Santa Rosa, CA, USA) with an initial denaturation at 95 °C for 4 min, followed by 35 cycles at 95 °C for 1 min, 58 °C for 1 min and 72 °C for 2 min and a final step at 72 °C for 10 min. The amplicons were verified by electrophoresis in a 1.5% agarose gel (Seakem, Invitrogen, Carlsbad, CA, USA) and stained with 10,000× GelRed (Biotium, Fremont, CA, USA).

All PCR products were cleaned with ExoSAP-IT (Affymetrix, Santa Clara, CA, USA), and both strands were individually sequenced using the BigDye Terminator v3.1 Cycle Sequencing Kit (Applied Biosystems, Carlsbad, CA, USA) in a 3130 Genetic Analyzer Sequencer (Applied Biosystems, Carlsbad, CA, USA) at Postgraduate College Facilities, Mexico, according to Juárez-Vázquez et al. [26].

### 2.4. Phylogenetic Analysis

The sequences of both DNA strands obtained in this study were assembled and trimmed with BioEdit v7.0.5 software [27] to create consensus sequences for each strain. These sequences were compared with the sequences deposited in the GenBank database of the National Center for Biotechnology Information (NCBI) (https://www.ncbi.nlm.nih.gov/) with the BLASTN v2.2.19 algorithm [28]. Reference and related sequences of *Trichoderma* spp. were downloaded from the GenBank database and included along with the sequences obtained in this study (MK778890, MK779008, MK791646, MK791647, MK791650, MK780094, MK779064, MK784067, MK791649, and MK791648). All consensus sequences were compiled into a single file, and then multiple sequences were aligned using Muscle [29] included in MEGA 7 software [30].

Phylogenetic reconstruction was performed with Bayesian inference using MrBayes v3.2.2 [31] with the evolutionary model GTR + gamma + invariant positions [32]. According to the instructions, the first 25% of the generated trees was discarded as the burn-in phase in each analysis, and posterior probabilities were determined for the remaining trees. The phylogenetic tree was rooted with *Hypomyces corticiicola* K. Põldmaa strain CBS 137.71 sequence. The standard deviation of split frequencies was stopped when the value was <0.01. The final tree was viewed with FigTree v1.4.4 software (http://tree.bio.ed.ac.uk/software/figtree/), and the sequences were deposited (Table 1) in the GenBank database (https://www.ncbi.nlm.nih.gov/genbank/).

Additionally, the *Phytophthora* isolated in this study was named here as *P. cinnamomi* strain CPO-PCU (GenBank accession number JQ266267). It was identified by the BLASTn algorithm and the distance tree of the result as FJ746646) deposited by the American Type Culture Collection with 100% maximum identity.

### 2.5. Morphological Characterization of Trichoderma spp.

Isolates of *Trichoderma* spp. were cultured on PDA medium and incubated at 25 °C for 5 days to determine their characteristics according to the taxonomic keys of Barnett and Hunter [33]. Later, discs of PDA (5 mm in diameter) of each strain were transferred according to Harris’ [34] modified method, in which microcultures were placed on a sterile rod triangle inside a Petri plate and incubated at 25 °C for 12 h and 24 h in the dark.

The macro and microscopic characteristics of *Trichoderma* spp. assessed were the presence of colony color, texture, presence of aerial mycelium and concentric rings, mycelium appearance on the upper and reverse sides, conidia shape and size (length and width), and morphology of 50 phialides were scored [15,35,36,37,38]. All *Trichoderma* spp. structures were observed under an optical microscope (Zeiss Axioskop 2 plus, Göttingen, Germany) at 40× and 100× magnification.

### 2.6. Antagonistic In Vitro Assessment of Trichoderma spp.

The variables considered for this analysis were the percentage of inhibition of radial growth (PIRG) and colonization with Bell’s scale [39]. For the in vitro test, *T. asperellum* (T-AS1, T-AS2, T-AS6, and T-AS7), *T. harzianum* (T-H3, T-H4, and T-H5), *T. hamatum* (T-A12), and *T. koningiopsis* (T-K8 and T-K11) were considered for dual confrontations versus *P. cinnamomi* in a completely randomized experimental design with 10 treatments and four repetitions. PDA discs (5 mm in diameter) with mycelia of *Trichoderma* spp. and *P. cinnamomi* were placed at the extremes of Petri plates containing PDA and incubated at 28 °C for 72 h. Then, mycelial growth was scored every 12 h until the first contact between the mycelia of each antagonist *T. harzianum*, *T. asperellum*, *T. koningiopsis*, and *T. hamatum* with the oomycete *P. cinnamomi* occurred [15,40,41].

Antagonism and percentage of inhibition were evaluated considering the mycelial growth radius of *Trichoderma* spp. and *P. cinnamomi* (with its respective control). Invasion of the antagonist or colonization on the surface of the *P. cinnamomi* mycelium was taken as the index of antagonism with the scale proposed by Bell [39] (Table 2).

The PIRG was calculated based on the formula proposed by Ezziyyani [41] (Equation (1)).
PIRG% = (R1 − R2)/R1 × 100(1)
where PIRG = Percent inhibition of radial growth; R1 = Radial growth (mm) of *P. cinnamomi* without *Trichoderma* spp; R2 = Radial growth (mm) of *P. cinnamomi* with *Trichoderma* spp.

### 2.7. Mycoparasitism and Colonization by Trichoderma spp. over P. cinnamomi

The colonization ability of *Trichoderma* spp. strains was evaluated by a dual-culture technique [42] that consisted of measuring the mycelial overgrowth of *Trichoderma* sp. every 12 h for 8 days from when the first contact occurred until the pathogen *P. cinnamomi* was completely covered (Figure 2). To calculate the percentage of *Trichoderma* colonization above *P. cinnamomi*, the formula proposed by Camporota [41] was used (Equation (2)).
C% = (DTP/DE) × 100(2)
where C% = The percentage of colonization; DTP = The distance of the route through which the colony of *Trichoderma* spp. grows over the pathogen *Phytophthora cinnamomi* colony, taking into account the axis that separates both colonies; DE = The distance between both colonies.

As a described mechanism of action of *Trichoderma* spp., macroscopic observations were made on the dual-cultures, considering invasion of the antagonist on the surface of the pathogenic mycelium as the index of mycoparasitism. The mycoparasitism was determined by microscopic observation on a Zeiss Axioskop plus (Zeiss, Göttingen, Germany) at 40× and 100× magnifications; the interactions of the antagonistic hyphae with those of the pathogen, either by coiling, vacuolation or penetration were documented.

Individual growth radii of the *Trichoderma* spp. antagonists and oomycete *P. cinnamomi* were measured by using a Vernier every 12 h until the Petri plate containing PDA was completely overgrown.

### 2.8. Fungal Growth and Plant Inoculation

*T. asperellum* (T-AS1, T-AS2, T-AS6, and T-AS7), *T. harzianum* (T-H3, T-H4, and T-H5), *T. koningiopsis* (T-K8 and T-K11), and *T. hamatum* (T-A12) were grown on PDA and incubated at 25 °C for 5 days. Later, they were exposed for 3 days to white light to induce the formation of conidia. Finally, conidia were harvested by filtering through a sterile cheesecloth [42]. Then, the conidial suspension was adjusted to a final concentration of 10^6^ conidia mL^−1^ using a hemocytometer (Marienfeld, Lauda-Königshofen, Germany).

The oomycete *P. cinnamomi* was incubated on PDA for 7 days. For the production of sporangia of *P. cinnamomi*, agar discs (5 mm in diameter) with mycelia were placed into 500-mL flasks with V8 juice liquid medium [5] and incubated at 28 °C for 8 days. Subsequently, the flasks were placed at 4 °C for 5 min; after that, the number of zoospores per milliliter was quantified with a hemocytometer. Finally, the zoospores were collected and adjusted to a final concentration of 10^6^ zoospores mL^−1^.

### 2.9. Avocado Seedling Inoculation

A greenhouse experiment was carried out to evaluate the antagonistic ability of 10 *Trichoderma* isolates recovered from avocado and cinnamon roots. For the experiment, the substrate was prepared with a mixture of peat moss, agricultural soil and vermiculite (2:2:1) sterilized with wet steam at 15 psi for 4 h and placed into 1 L plastic pot. Avocado seeds var. *drymifolia* were collected from trees in the municipality of Atlixco in Puebla, Mexico, which has been recognized as having moderate resistance to infection by *P. cinnamomi* [5]. They were disinfested with 3% sodium hypochlorite for 10 min, rinsed with sterile distilled water, and planted in plastic pots. 

A total of 12 treatments, with 4 replicates per treatment (Table 1) were distributed in the greenhouse using a complete randomized design. Inoculations of 20–60 day old avocado seedlings (15 to 20 cm high) with *Trichoderma* spp. and *P. cinnamomi* were performed by adding 5 mL of 10^6^ conidia mL^−1^ and 10^6^ zoospores mL^−1^ to the substrate, while control seedlings were inoculated with sterile distilled water; an additional control was *P. cinnamomi* alone. The avocado seedlings were maintained in the greenhouse for three months.

The disease incidence (Ii), based on visual symptoms such as the number of asymptomatic seedlings and the appearance of wilt symptoms and dead seedlings was evaluated at 20 days after inoculation (dai) with the following equation (Equation (3)):Ii% =∑ (ni/Ni) × 100(3)
where Ii = Incidence of diseased seedlings at time i; ni = Number of diseased seedlings at time i; Ni = Total population of inoculated seedlings.

### 2.10. Statistical Analysis

The PIRG, in vitro colonization percentage (C%) and in vivo incidence of disease (Ii%), were expressed as percentages and transformed with angular arccosine √x + 1. The mean values of the four repetitions (five plants per repetition) of the biological samples were subjected to an analysis of variance with Statistical Analysis System software SAS [43], using the Tukey test to determine the significant differences between the treatments and level of significance of *p* < 0.05.

## 3. Results

### 3.1. Isolates Recovered

Ten *Trichoderma* spp. and one *Phytophthora* isolate were recovered from avocado and cinnamon roots from five states in Mexico. They were studied to establish their identity based on phylogenetic and morphological characteristics. Additionally, to determine the *Trichoderma* spp. biocontrol activity over the oomycete *Phytophthora* sp. in vitro and under greenhouse conditions.

### 3.2. Phylogenetic Analysis

This analysis was performed with sequences of a 621 bp fragment of the ITS region corresponding to *Trichoderma* spp. and 849 bp of *P. cinnamomi*. The isolates in study were clustered within four subclades belonging to *T. asperellum* (T-AS1, T-AS2, T-AS6, and T-AS7), *T. hamatum* (T-A12), *T. harzianum* (T-H3, T-H4, and T-H5), and *T. koningiopsis* (T-K8 and T-K11). A phylogenetic tree constructed with Bayesian inference was run until 2 million generations with a final standard deviation of 0.008126.

The evolutionary history of the indigenous *Trichoderma* species represented in the phylogenetic tree (Figure 3) shows that they represented four species, which are part of the soil microorganism community in the avocado ecological niche. Furthermore, these species can coexist in the same root area of the same avocado tree of the *drymifolia* variety.

### 3.3. Morphological Characterization of Trichoderma spp.

The results indicated the presence of 10 *Trichoderma* strains that belong to four species. They were characterized by the presence of a light to dark green colony color, aerial mycelium, light yellow staining of the PDA, and one to three concentric rings (Figure 4A,D,G,J and Appendix A).

Strains of *T. asperellum* were characterized by the absence of color on the reverse side of the Petri plate. *T. harzianum* presented light to dark green colonies with a dusty-cottony texture and aerial mycelia (Figure 4A), *T. hamatum* showed yellow pigment on the reverse side of the Petri plate, and *T. koningiopsis* presented flocculated aerial mycelia that adhered to the culture medium (Figure 4G).

The conidia of *T. harzianum* were subglobose, with elongated globus phyloids distributed asymmetrically (Figure 4B,C). *T. asperellum* presented bottle-shaped phialides with a length of 10.2 to 11.8 μm and arranged in spiral shapes of 2, 3, and 4, and the conidia were globose to subglobose and occasionally ovoid (Appendix A). The chlamydospores in *T. asperellum* appeared at the end of the mycelium with a globose to subglobose form (Figure 4K,L); the morphological characteristics of the colonies of *Trichoderma* spp. isolated in this study are already mentioned above, described in Appendix A.

### 3.4. Percentage of Growth Inhibition In Vitro

In the dual-assays, some *Trichoderma* spp. could compete for space and inhibit the growth of *P. cinnamomi* (Table 3). Double confrontation between *T. asperellum* (T-AS1 and T-AS2) and *T. harzianum* (T-H3 and T-H4 strains) showed an inhibition above 65.3% (*p* < 0.05). However, the highest percentage of inhibition, 78.32%, was obtained with *T. asperellum* (T-AS2 strain). Nevertheless, according to the results, these strains have the potential to inhibit the mycelial growth of the *P. cinnamomi* strain CPO-CPU. In the same way, *T. harzianum* strain T-H3 presented 73.33% inhibition (Table 3). The data analysis showed that mycelial growth inhibition of *P. cinnamomi* by *Trichoderma* species had significant differences (*p* < 0.05).

The lowest percentage of inhibition was observed for *T. harzianum* strain T-H5 with 38.66%. In addition, T-H4 and T-AS1 strains belonging to *T. harzianum* and *T. asperellum*, respectively, presented moderate inhibition of 69.15%. The species with low to moderate performance were *T. asperellum* T-AS7 and T-AS6 strains, with 51.5% and 53.33%, respectively; *T. hamatum* strain T-A12 with 50.99%; *T. koningiopsis* T-K8 and T-K11 strains with 46.99% and 47.33%, respectively (Table 3).

### 3.5. Mycoparasitism, Percentage of Colonization, and Growth Rate

In the dual confrontation test, the 10 *Trichoderma* strains had the ability to colonize the *P. cinnamomi* strain CPO-CPU, reducing mycelial growth (Table 4). The best colonization percentage was observed with *T. koningiopsis* T-K11 (67.83%) and *T. harzianum* T-H3 (60%) strains. Generally, most of the *Trichoderma* showed significant differences (*p* < 0.05) and they reduced the mycelial growth of the *P. cinnamomi* strain CPO-CPU. However, the strains of *T. asperellum* (T-AS2, T-AS6, and T-AS7) and *T. koningiopsis* (T-K8) presented intermediate colonization. 

The results showed that the inhibition and colonization of all *Trichoderma* strains were efficient considering the competition for space and nutrients. In addition, the parasitic ability against the *P. cinnamomi* strain CPO-CPU was observed by light microscopy and corroborated in microcultures (Figure 5A–D). Additionally, in the dual-assays, which were performed by facing up the indigenous strains of *Trichoderma* spp. and *P. cinnamomi* strain CPO-CPU, there were different types of mycoparasitism as vacuolation, surrounding and lysing the mycelium of the pathogen and abnormal mycelium, and massive hyphal curl of *T. hamatum* and *T. koningiopsis* covering the pathogen *P. cinnamomi* (Figure 5A–D).

According to Bell’s scale [40] (Table 2 and Table 3), strains of the *Trichoderma* species were classified into two types: In scale 1, when biocontrol completely covers the surface, it should be occupied by the pathogen. In this case, *T. koningiopsis* (T-K8 and T-K11), *T. harzianum* (T-H3), and *T. asperellum* (T-AS6) were overgrown on the *P. cinnamomi* strain CPO-CPU. In scale 2, when *Trichoderma* overgrew at least two-thirds of the medium surface, it should be occupied by the pathogen. For this case, *T. harzianum* (T-H4 and T-H5), *T. hamatum* (T-A12), and *T. asperellum* (T-AS1) strains presented a colonization value between 23.83% and 30% on the *P. cinnamomi* strain CPO-CPU (Table 3).

In the dual confrontation test, the 10 *Trichoderma* strains had the ability to colonize the *P. cinnamomi* strain CPO-CPU, reducing mycelial growth. The best colonization percentage was observed with *T. koningiopsis* T-K11 (67.83%) and *T. harzianum* T-H3 (60%) strains. Generally, most of the *Trichoderma* showed significant differences (*p* < 0.05) reducing the mycelial growth of the *P. cinnamomi* strain CPO-CPU. However, the strains of *T. asperellum* (T-AS2, T-AS6, and T-AS7) and *T. koningiopsis* (T-K8) presented intermediate colonization. The lowest colonization values were observed with *T. hamatum* (T-A12) and *T. harzianum* (T-H4) strains with 27.33% and 30% colonization, respectively (Table 3).

The *P. cinnamomi* strain CPO-CPU had uniform radial growth with no significant differences in the growth rate, while the *Trichoderma* strains showed significant differences (*p* < 0.05) related to the individual growth rate. The strains that showed the best development rates were *T. koningiopsis* (T-K8) and *T. hamatum* (T-A12); otherwise, the lowest levels were presented by *T. harzianum* (T-H5) and *T. asperellum* (T-AS2, T-AS6, and T-AS7), with similar behaviors (Table 4). These results indicated the antagonistic capacity of *Trichoderma* species in controlling the *P. cinnamomi* strain CPO-CPU growth under in vitro conditions.

### 3.6. Biocontrol of P. cinnamomi by Trichoderma Strains in the Greenhouse Assay

Avocado seedlings inoculated in the greenhouse with both *Trichoderma* and *P. cinnamomi* strain CPO-CPU showed the first visible symptoms at 20 days. The disease incidence (Ii) caused by *P. cinnamomi* leads to a high mortality rate of avocado seedlings; although, it was observed that the antagonistic effect of the different isolates of *Trichoderma* spp. reduced the absence of symptoms of wilt or root rot in relation to the seedling control group (Appendix A), showing significant differences (*p* < 0.05).

When the antagonists inoculated avocado seedlings previously infected with *P. cinnamomi* strain CPO-CPU, there were significant differences (*p* < 0.05) due to the absence of wilt or root rot symptoms in relation to the seedling control. The inoculation of avocado seedlings with *T. asperellum* (T-AS2 and T-AS7) resulted in only 5% death by root rot caused by *P. cinnamomi* strain CPO-CPU (Figure 6).

The disease incidence of seedlings with wilt symptoms and death caused by *P. cinnamomi* strain CPO-CPU decreased more than 50% in the presence of *Trichoderma* spp. The strain with the best behavior for controlling this pathogenic oomycete was *T. asperellum* (T-AS7), with 80% of seedlings without evident symptoms (Figure 6).

## 4. Discussion

In this work, we have demonstrated that indigenous endophytic *Trichoderma* spp. recovered from avocado and cinnamon roots were efficient in vitro and under greenhouse conditions as biological control agents against the soil-borne pathogen *P. cinnamomi*, reducing the incidence of root rot in avocado seedlings.

Species of the genus *Trichoderma*, well known as green-spored fungi, have been reported widely as biological control agents against plant diseases caused mainly by soil-borne pathogens [44,45]. They are filamentous fungi that exhibit different modes of action such as growth regulators, competition by space and nutrients, and infection sites and reduce the damage by pathogens that affect the development of the roots [8,17,46]. Additionally, the *Trichoderma* genome has revealed the presence of genes involved in a mutualistic association between plants and biological control agents, in which some species of this genus can be established in the rhizosphere of the plant [47].

In this study, *Trichoderma* species, such as *T. asperellum*, *T. hamatum*, *T. harzianum*, and *T. koningiopsis,* were clearly identified based on phylogenetic and morphological approaches and have been frequently isolated from the rhizosphere worldwide [38,48]. Phylogenetic reconstruction did not show any discrepancy with the morphological features found in this work, and phylogenetic studies using the ITS region were able to classify each strain within the corresponding species.

The four species of *Trichoderma* identified in this study are inhabitants of different regions and tropical systems around the world [46,49]. Their ecological plasticity is related to the capacity to produce enzymes to degrade substrates with a high organic matter content found in soil, rhizosphere, leaf litter, wood, aquatic systems, and less frequently in biological tissues [36,50]. The strains belonging to *T. harzianum* have a greater capacity than the soil-borne plant pathogen to compete for root space [16,41]. Similar behavior was observed in *T. asperellum* [17,48], *T. koningiopsis*, and *T. hamatum*, which act as antagonistic agents during root colonization [49,51]. However, this last species has not yet been isolated from cinnamon and avocado roots to date. It can be indicative that they have evolved with the host to settle in the rhizosphere or live like an endophytic fungus inside the roots of these trees.

The endophytic capacity and ability of a certain group of individuals to create a strengthened relationship between the host and other microbial species, allowing them to coexist like an ecological unit, is known as a halobiont [52]. This feature explains the recasts in the complex association between *Trichoderma* spp. with avocado roots [15,53].

Antagonistic activity of these *Trichoderma* spp. evaluated with Bell’s scale [39] showed that most strains were scored in class 1, with *Trichoderma* completely overgrown on *P. cinnamomi* strain CPO-CPU, covering the entire surface of the culture medium. Occasionally, some *Trichoderma* strains were in class 2, covering at least two-thirds of culture medium surface, suggesting the importance of these strains as potential biological control agents.

Additionally, when healthy avocado seedlings with the absence of symptoms were inoculated with *T. harzianum* (T-H5), *T. asperellum* (T-AS2), and *T. koningiopsis* (T-K11), they presented a lower Ii of root rot or wilt symptoms along with *T. asperellum* (T-AS7). In the case of *T. harzianum* (T-H4 and T-AS6) that exhibited wilting symptoms and dead seedlings by root rot, the results indicated highly significant differences concerning the growth of the *P. cinnamomi* strain CPO-CPU. The comparison performed with the control seedlings inoculated only with this oomycete revealed symptoms of root rot, growth retardation, wilting, defoliation, root necrosis, and seedling death. In contrast, in the control (without an inoculum) avocado seedlings, these symptoms did not occur, and seedling development was considered normal. Similar results were obtained in studies focused on the activity of *Trichoderma* in roots from other crops [54,55].

Furthermore, these results reflect that the strains of *Trichoderma* spp. act as antagonists and play significant roles in plant growth, and others display an efficient mycoparasitism mechanism against soil-borne plant pathogens, including competition for space within the ecological niche, appressorium formation, and antibiosis. In this response, plants activate induced systemic resistance, a process that has been well documented [44,46,56]. Observations made in vitro between the inter-action points of the plant pathogen *P. cinnamomi* and *Trichoderma* spp. were supported by microscopic observations of mycoparasitism, in which vacuolation in the mycelium of the oomycete, disintegration, and parallel mycelial growth were found [46]. The colonization capacity of the biological control agents is considered essential to establish themselves in the rhizosphere, especially in root endophytes or those inhabiting the ecological niche of the avocado and cinnamon roots.

The mechanisms of action of *Trichoderma* species prevent the host from being affected by other kinds of pathogens. In the case of *T. asperellum*, it contributes to the active mechanisms against biotic and abiotic stress (salinity, low availability of nutrients in the soil, and root infection), conferring resistance to the plant in the presence of soil-borne pathogens [57]. Another interesting aspect of *T. koningiopsis* is the release of koninginol, the compounds isolated from the endophytic fungus *T. koningiopsis* (10S-7-isopropyl-4,10-dimethylbicyclo), which has potent antimicrobial activity, preventing infection by bacteria and fungi spread through the soil [58].

## 5. Conclusions

This study reports to *T. asperellum*, *T. hamatum*, *T. harzianum*, and *T. koningiopsis* isolated from avocado and cinnamon roots, which have been very poorly studied in Mexico in those hosts. Most species of *Trichoderma* isolated in this work partially or totally covered oomycete growth, reducing the incidence rate of root rot, wilting, and death of avocado seedlings in the greenhouse. On the other hand, inoculation with *Trichoderma* spp. showed that the percentage of *P. cinnamomi* infection severity decreased, showing an increase in the health of avocado seedlings by the reduction in the percentage of wilting, root rot, and death of plants. The results of this study provide evidence that endophytic indigenous species of *Trichoderma* isolated from avocado and cinnamon roots efficiently contributed to in vitro and greenhouse biocontrol against the plant pathogen oomycete *P. cinnamomi*, the causal agent of root rot of avocado worldwide.

## Figures and Tables

**Figure 1 plants-09-01220-f001:**
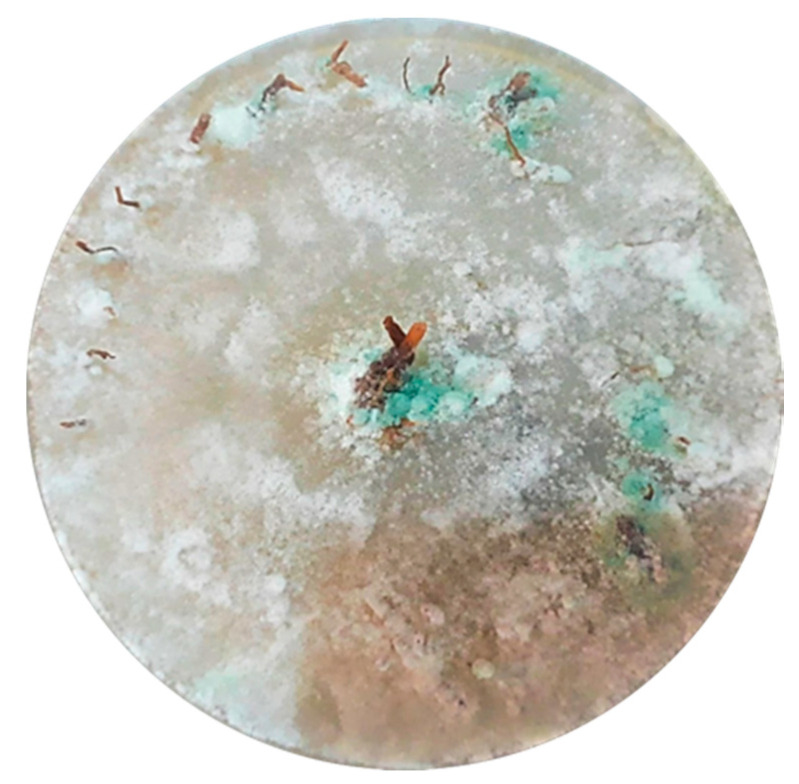
Mycelial growth of *Trichoderma* spp. and green sporulation on avocado and cinnamon roots on PDA medium.

**Figure 2 plants-09-01220-f002:**
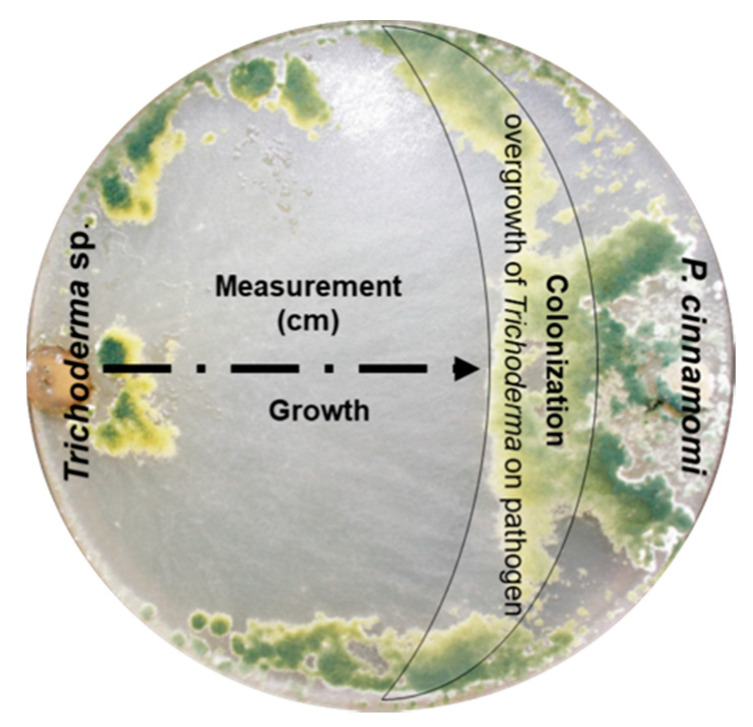
Aspect of *Trichoderma* strain colonization over the plant pathogen *Phytophthora cinnamomi* after 8 days.

**Figure 3 plants-09-01220-f003:**
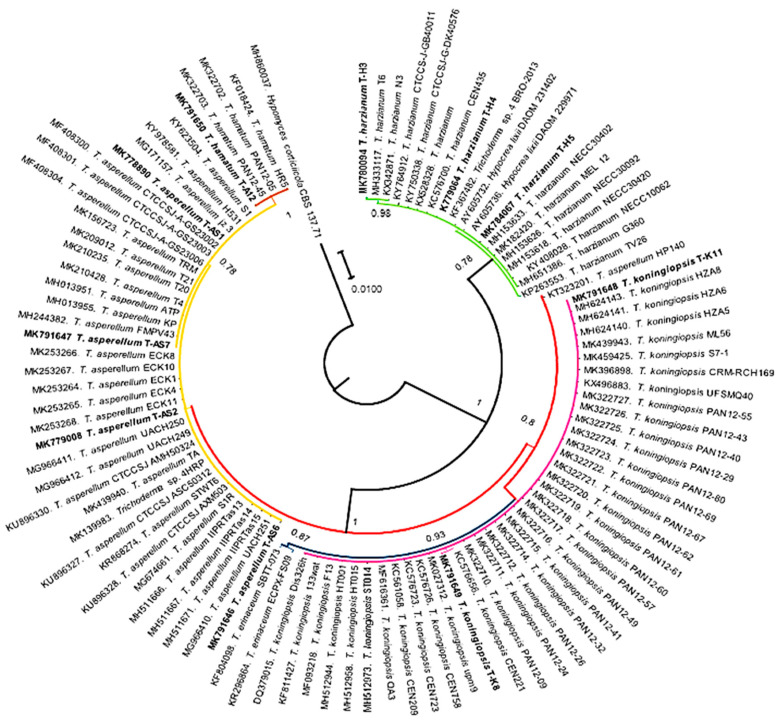
Phylogenetic tree constructed with sequences corresponding to the internal transcribed spacer region of indigenous species of *Trichoderma* and related sequences. The Bayesian inference method using the General Time Reversible substitution model was implemented with two million generations with a final standard deviation of <0.01. *Hypomyces corticiicola* was considered as an outgroup.

**Figure 4 plants-09-01220-f004:**
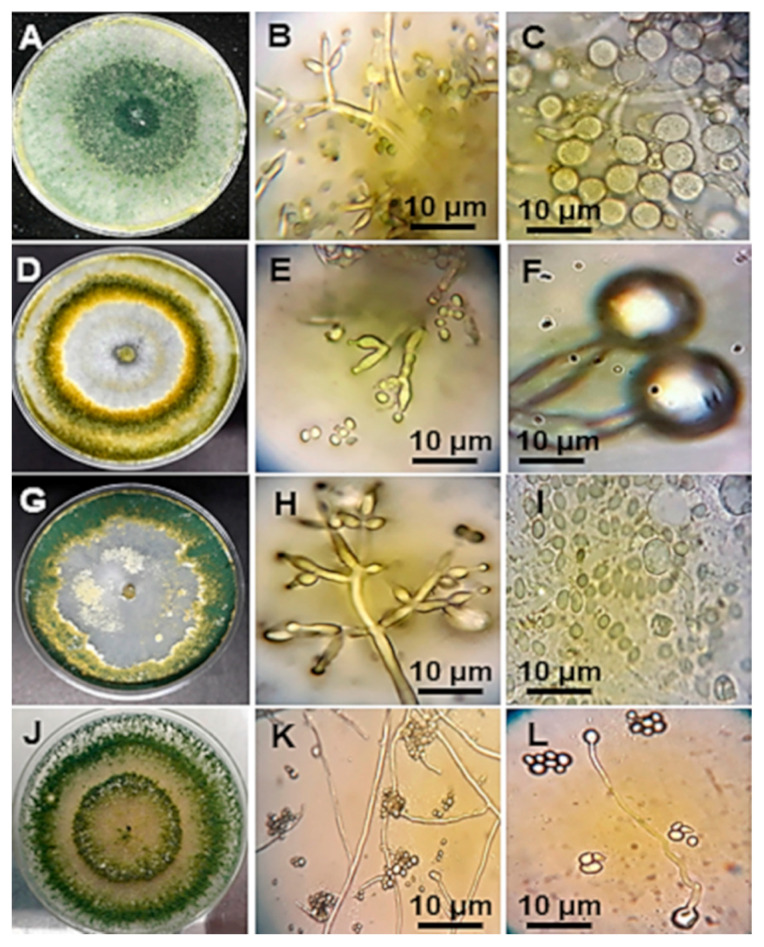
Macro and microscopic characteristics of the indigenous endophytic *Trichoderma harzianum* (**A**–**C**), *T. hamatum* (**D**–**F**), *T. koningiopsis* (**G**–**I**) and *T. asperellum* (**J**–**L**) isolated from avocado and cinnamon root sections. (**A**) Colony showing dark green color without the presence of rings. (**B**) Presence of phialides and conidia. (**C**) Globose chlamydospores and germ tubes. (**D**) Colony with dark green tone and yellow color, with abundant mycelia between the two to three bright rings. (**E**) Phialides with conidia. (**F**) Germinating chlamydospores. (**G**) Light to dark green colony with one and two rings. (**H**) Conidiophore- and bottle-shaped phialides. (**I**) Conidia and chlamydospores with germination tubes. (**J**) Light green to dark colony, with a ring and abundant mycelia. (**K**) Slightly balloon like phialides. (**L**) Dehydrated conidia germinating with a thin mycelium and terminal chlamydospore.

**Figure 5 plants-09-01220-f005:**
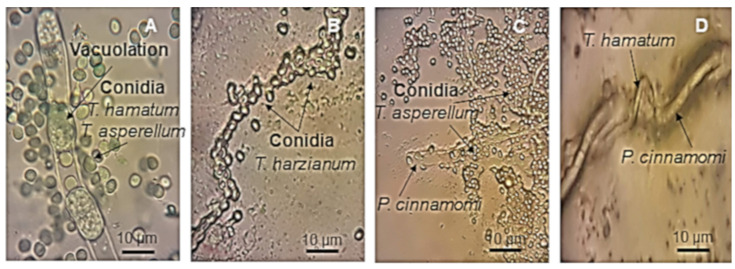
Hyphal interactions and mycoparasitism of indigenous species of *Trichoderma* spp. observed at 100× magnification. (**A**) Vacuolation of *Phytophthora cinnamomi* strain CPO-CPU induced by *Trichoderma hamatum* and *T. asperellum*. (**B**) Conidia of *T. harzianum* surrounding and lysing the mycelium of the pathogen. (**C**) Conidia of *T. harzianum* and *T. asperellum* surrounding the mycelium of *P. cinnamomi*. (**D**) Abnormal mycelium and massive hyphal curl of *T. hamatum* and *T. koningiopsis* covering the pathogen *P. cinnamomi*.

**Figure 6 plants-09-01220-f006:**
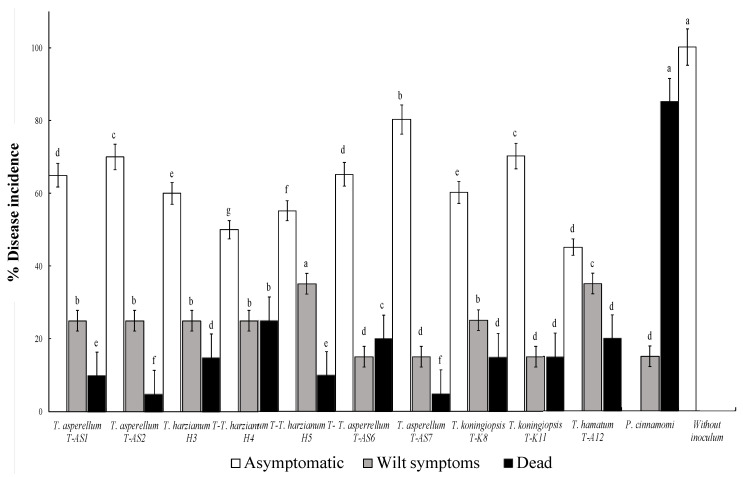
Avocado seedling interactions of 10 indigenous species of *Trichoderma* spp. inoculated individually with *Phytophthora cinnamomi*. The percentage of disease incidence is reduced under greenhouse conditions. Different letters denote the treatments that are significantly different according to Tukey’s test at *p* < 0.05.

**Table 1 plants-09-01220-t001:** Endophytic species of *Trichoderma* and *Phytophthora cinnamomi* isolated from avocado and cinnamon roots recovered from different states of Mexico.

Strain	*Trichoderma* Species ^a^	Host	GenBank Accession Number	Location
T-AS1	*T. asperellum*	*Persea americana*	MK778890	Hidalgo
T-AS2	*T. asperellum*	*Persea americana*	MK779008	Hidalgo
T-H3	*T. harzianum*	*Persea americana*	MK780094	Tepeyanco, Tlaxcala
T-H4	*T. harzianum*	*Persea americana*	MK779064	Tetela de Ocampo, Puebla
T-H5	*T. harzianum*	*Persea americana*	MK784067	La Aurora, Tepeyanco, Tlaxcala
T-AS6	*T. asperellum*	*Persea americana*	MK791646	Uruapan, Michoacan
T-AS7	*T. asperellum*	*Persea americana*	MK791647	Uruapan, Michoacan
T-K11	*T. koningiopsis*	*Persea americana*	MK791648	Jalisco, Guadalajara
T-K8	*T. koningiopsis*	*Persea americana*	MK791649	Atlixco, Puebla
T-A12	*T. hamatum*	*Cinnamomun verum*	MK791650	Tlaltipa, Zozocolco de Hidalgo, Veracruz
CPO-PCU ^b^	*Phytophthora cinnamomi*	*Persea americana*	JQ266267	Uruapan, Michoacan

^a^ Identification based on a phylogenetic reconstruction analysis with Bayesian inference. ^b^ CPO-PCU means Postgraduate College in Agricultural Science (CPO) and strain designation (PCU).

**Table 2 plants-09-01220-t002:** *Trichoderma* strain antagonism evaluated in vitro using Bell’s scale [39], considering the invasion of the surface.

Scale	Antagonistic Ability
1	*Trichoderma* completely overgrew the *P. cinnamomi* and covered the entire medium surface.
2	*Trichoderma* overgrew at least two-thirds of the medium surface.
3	*Trichoderma* and the *P. cinnamomi* each colonized approximately one-half of the medium surface and neither organism appeared to dominate the other.
4	*P. cinnamomi* colonized at least two-thirds of the medium surface and appeared to withstand encroachment by *Trichoderma*.
5	*P. cinnamomi* completely overgrew the *Trichoderma* and occupied the entire medium surface.

**Table 3 plants-09-01220-t003:** Evaluation of the percentage of inhibition and colonization of *Trichoderma* spp. over *Phytophthora cinnamomi*.

Strain	*Trichoderma* Species	PIRG		Standard	Colonization		Standard
		%		Error	%		Error
T-AS2	*T. asperellum*	78.32	^a^	5.03	44.33	^c^	4.66
T-H3	*T. harzianum*	73.33	^ab^	2.87	60.00	^ab^	1.63
T-H4	*T. harzianum*	69.15	^b^	1.13	30.00	^d^	1.27
T-AS1	*T. asperellum*	65.30	^b^	1.51	29.66	^d^	3.28
T-AS6	*T. asperellum*	53.33	^c^	2.84	51.00	^bc^	2.84
T-AS7	*T. asperellum*	51.50	^c^	6.06	46.83	^c^	1.10
T-A12	*T. hamatum*	50.99	^c^	6.65	27.33	^d^	2.21
T-K11	*T. koningiopsis*	47.33	^cd^	4.55	67.83	^a^	2.47
T-K8	*T. koningiopsis*	46.99	^cd^	1.27	55.00	^bc^	2.20
T-H5	*T. harzianum*	38.66	^d^	5.10	23.83	^d^	3.53

Means followed by the same letter are not significantly different for *p* ≤ 0.05 according to Tukey test.

**Table 4 plants-09-01220-t004:** Test of means of the growth rate of *Trichoderma* spp. and *Phytophthora cinnamomi*.

Strain	*Trichoderma* Species	Growth Rate			
		*Trichoderma* spp.		*P. cinnamomi*	
T-K8	*T. koningiopsis*	0.1176	^a^	0.0526	^a^
T-A12	*T. hamatum*	0.1132	^ab^	0.0462	^a^
T-H3	*T. harzianum*	0.0985	^abc^	0.0098	^a^
T-H4	*T. harzianum*	0.091	^abcd^	0.0384	^a^
T-AS1	*T. asperellum*	0.0346	^bcde^	0.0379	^a^
T-AS2	*T. asperellum*	0.0295	^cde^	0.0342	^a^
T-AS7	*T. asperellum*	0.0212	^cde^	0.0169	^a^
T-K11	*T. koningiopsis*	0.0285	^cde^	0.0512	^a^
T-AS6	*T. asperellum*	0.0174	^de^	0.0240	^a^
T-H5	*T. harzianum*	0.0077	^e^	0.0316	^a^

Means followed by the same letter are not significantly different for *p* ≤ 0.05 according to Tukey test.

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
