# Peer review of "Endophytic Trichoderma Species Isolated from Persea americana and Cinnamomum verum Roots Reduce Symptoms Caused by Phytophthora cinnamomi in Avocado"

_plants, 2020, doi:10.3390/plants9091220_

Round 1
Reviewer 1 Report
This work constitutes a valuable contribution to the management of Avocado root rot using Trichoderma spp. The introduction would benefit from proof-reading as there are multiple errors throughout the text. The methods are adequate, well described and reproducible, but some additional details are needed regarding the collection sites and the outcome of a normality test to support the choice of statistical tests. The results are, in general, well described. The discussion would benefit from additional comparison between the results of this study and those obtained using Trichoderma in other systems. Given the above, my recommendation is to accept after minor revision (and proof-reading). Some detailed feedback is provided below:
Please italicize scientific names in the title
Abstract
Line 20. “biocontrol capacity of Trichoderma spp. strains against… ”
Line 24. What was does the reported “value” correspond to? More detail about the methods is needed in the abstract.
Line 26-28: Measured incidence values? Were these significantly lower?
Introduction
Line 33. “fruits”
Line 35. Reference?
Line 43. “show typical rot symptoms”
Line 44. “a mortality of up to 100%”
Line 47. Use “chemical fungicides” instead of “chemical compounds”
Line 50. Use “offer sustainable alternatives” instead of “has become the best alternative”
Line 54. Use “identified” instead of “studied”
Line 55. Please rephrase b) to “acting as natural mycoparasites and antagonistic agents”
Line 61. Be consistent with the italization of “in vitro”
Line 64. “are non-existent” or “do not exist”
Line 65. Use “most successful” instead of “most proven”
Line 75. Use “tested” instead of “evaluated”
The text requires extensive language revision. I will not continue providing detailed feedback for the rest of the text but strongly advice the authors to engage a proof-reader.
Materials and methods
2.1. Please provide detailed information about the collection sites (geographical coordinates) and sampling methods (e.g., distance between sampling sites).
2.2. Please describe the morphological characteristics of Trichoderma and Phytopthora
Line 171. Add space after table 2.
Line 210. “10 Trichoderma isolates recovered from avocado and cinnamon roots”
Line 231. Please provide the results of a normality test to prove that transformation was successful. A homogeneity of variances test would be advisable two as these assumptions must be met for ANOVA.
Line 240. Italicize Phytophthora
Line 240. “greenhouse conditions”
Line 266. Italicize T. asperellum
Line 334. Italicize T. koningiopsis
Line 342. italicize Trichoderma
In figure 6. It is unclear which statistical test was used. Was this an ANOVA for each symptomatic category (Asymptomatic, wilt symptoms, dead)? If this is the case, please explain so clearly in the legend.
Discussion
Please include additional comparisons between your results and those of other studies using Trichoderma (in other systems).
Author Response
Response to Reviewer 1 Comments
This work constitutes a valuable contribution to the management of Avocado root rot using Trichoderma spp. The introduction would benefit from proof-reading as there are multiple errors throughout the text. The methods are adequate, well described and reproducible, but some additional details are needed regarding the collection sites and the outcome of a normality test to support the choice of statistical tests. The results are, in general, well described. The discussion would benefit from additional comparison between the results of this study and those obtained using Trichoderma in other systems. Given the above, my recommendation is to accept after minor revision (and proof-reading). Some detailed feedback is provided below:
Point 1: Please italicize scientific names in the title
Response 1: As suggested, scientific names in the title were italicized.
Point 2: Abstract. Line 20. “biocontrol capacity of Trichoderma spp. strains against…”
Response 2: This sentence was re-write.
Point 3: Line 24. What was does the reported “value” correspond to? More detail about the methods is needed in the abstract.
Response 3: Line 28 and 29. ...In vitro dual-culture assay, the percentage of inhibition of radial growth (PIRG)...
Point 4: Line 26-28: Measured incidence values? Were these significantly lower?
Response 4: Line 32-36. In the greenhouse, the infection was evaluated according to the percentage of disease incidence. Plants with the lowest incidence of dead by avocado root rot were those whose seedlings were inoculated with T-AS2 and T-AS7 resulted in only 5% death by root rot caused by P. cinnamomi. The disease incidence of seedlings with wilt symptoms and death decreased more than 50% in the presence of Trichoderma spp....
Point 5: Introduction. Line 33. “fruits”
Response 5: The authors consider the word ‘fruit’ because in only a kind of fruit.
Point 6: Line 35. Reference?
Response 6: Line 43, reference [6].
Point 7: Line 43. “show typical rot symptoms”
Response 7: Line 51. ... roots rot symptom...
Point 8: Line 44. “a mortality of up to 100%”
Response 8: It was corrected according the suggestion given by reviewer 2.
Point 9: Line 47. Use “chemical fungicides” instead of “chemical compounds”
Response 9: The correction was incorporated.
Point 10: Line 50. Use “offer sustainable alternatives” instead of “has become the best alternative”
Response 10: The suggestion was added.
Point 11: Line 54. Use “identified” instead of “studied”
Response 11: It was corrected according the suggestion given by reviewer 2, described.
Point 12: Line 55. Please rephrase b) to “acting as natural mycoparasites and antagonistic agents”
Response 12: The suggestion was added.
Point 13: Line 61. Be consistent with the italization of “in vitro”
Response 13: The word “in vitro” was italicized in the full text.
Point 14: Line 64. “are non-existent” or “do not exist”
Response 14: The suggestion ‘do not exit’ was included.
Point 15: Line 65. Use “most successful” instead of “most proven”
Response 15: This paragraph was eliminated, suggestion given by reviewer 2
Point 16: Line 75. Use “tested” instead of “evaluated”
Response 16: The change was made.
Point 17: The text requires extensive language revision. I will not continue providing detailed feedback for the rest of the text but strongly advice the authors to engage a proof-reader.
Response 17: Manuscript was edited and re-edited by Springer Nature Author Service.
‘Thank you again for giving us the opportunity to address your concerns. You should have received an email that your re-edited file has been uploaded to your account, and your editing certificate has been updated with a new date.
Elizabeth Sung, PhD
Research Communication Partner
Springer Nature Author Services
SNAS Complete files 8FTGSVQ7
Point 18: Materials and methods.
2.1. Please provide detailed information about the collection sites (geographical coordinates) and sampling methods (e.g., distance between sampling sites).
Response 18: was incorporated, in Line 88 to 93.
Point 19: 2.2. Please describe the morphological characteristics of Trichoderma and Phytopthora
Response 19: Line 109-110 ... (hyphal swellings, chlamydospore and sporangium) and 186 – 188: ... The macro- and microscopic characteristics of Trichoderma spp. assessed were the presence of colony color, texture, presence of aerial mycelium and concentric rings, mycelium appearance on the upper and reverse sides, conidia shape and size (length and width), and morphology...
Point 20: Line 171. Add space after table 2.
Response 20: The space was added.
Point 21: Line 210. “10 Trichoderma isolates recovered from avocado and cinnamon roots”
Response 21: As suggested, the new sentence was incorporated.
Point 22: Line 231. Please provide the results of a normality test to prove that transformation was successful. A homogeneity of variances test would be advisable two as these assumptions must be met for ANOVA.
Response 22: In section 2.10, it is explained that the in vitro colonization percentage (C%) was expressed in percentages and transformed with the angular arccosine √x + 1. Subsequently, they were subjected to an analysis of variance and finally the test of Tukey to determine the significant differences between the treatments at a significance level of p <0.05, with the SAS software of the Statistical Analysis System, where the normality test of the data is already implicit.
Point 23: Line 240. Italicize Phytophthora
Response 23: It was corrected.
Point 24: Line 240. “greenhouse conditions”
Response 24: It was corrected.
Point 25: Line 266. Italicize T. asperellum
Response 25: It was corrected.
Point 26: Line 334. Italicize T. koningiopsis
Response 26: It was corrected.
Point 27: Line 342. italicize Trichoderma
Response 27: It was corrected.
Point 28: In figure 6. It is unclear which statistical test was used. Was this an ANOVA for each symptomatic category (Asymptomatic, wilt symptoms, dead)? If this is the case, please explain so clearly in the legend.
Response 28: It was made in the figure 6.
Point 29: Discussion. Please include additional comparisons between your results and those of other studies using Trichoderma (in other systems).
Response 29: review.
Reviewer 2 Report
The manuscript presents results on the suitability of the use of microbial antagonists for the control of avocado root rot. The approach based on the use of endophytes from the plant itself is interesting and enhances the adoption of conservative biological control methods.
The manuscript will be of great interest to those interested in this discipline and the crop in question.
In general, it is well prepared and presented, and a series of changes have been suggested regarding grammar and / or formal aspects indicated by hand directly on the manuscript.
In addition, the authors should take into account and correct (in case it is not an effect of the layout by the publisher) the appearance and proportionality of the figures provided.

Author Response
Response to Reviewer 2 Comments + PDF
The manuscript presents results on the suitability of the use of microbial antagonists for the control of avocado root rot. The approach based on the use of endophytes from the plant itself is interesting and enhances the adoption of conservative biological control methods.
The manuscript will be of great interest to those interested in this discipline and the crop in question.
In general, it is well prepared and presented, and a series of changes have been suggested regarding grammar and / or formal aspects indicated by hand directly on the manuscript.
In addition, the authors should take into account and correct (in case it is not an effect of the layout by the publisher) the appearance and proportionality of the figures provided.
All suggestions in the pdf file have been incorporated.
Point 1: Line 20: “endophytic” (line 23)
Response 1: It was corrected.
Point 2: Line 21 ...both in vitro... (line 23)
Response 2: It was inserted.
Point 3: Line 35: “global”
Response 3. Was corrected.
Point 4: Line 39: ...this some..
Response 4: Was corrected... This sentence was re-write.
Point 5: line 44: “that reaches up to” 100%..
Response 5: Was corrected...
Point 6: Line 49: ...“ practices”...
Response 6: Was corrected.
Point 7: Line 49: ...”control”...
Response 7: Was corrected.
Point 8: Line 54: “described”
Response 8: Was inserted
Point 9: Line 64: ...there are not...
Response 9: was inserted in Line 70: However, there are not experiments...
Point 10: Line 65: “Among the most proven in vitro biocontrol agents against soil-borne plant pathogens are Trichoderma spp., which inhibit the mycelial growth of phytopathogens and promote seedling growth”
Response 10: Line 65: was deleted.
Point 11: Line 72: ...”plants”...
Response 11: was inserted.
Point 12: Line 73: ... “some” indigenous...
Response 12: Was corrected.
Point 13: Line 74: ...”tested”in vitro...
Response 13: was corrected.
Point 14: Line 79: ... drymifolia...
Response 14: was corrected.
Point 15: Line 81: Cinnamomum verum J. Presl
Response 15: was corrected (binomial with authors)
Point 16: Line 91: ...”colonies with”...
Response 16: was corrected.
Point 17: Line 98: ... PDA...
Response 17: was corrected.
Point 18: Line 101: “Genomic” ...
Response 18: was corrected.
Point 19: Line 134: “K. Põldmaa”
Response 19: Was inserted.
Was inserted line 142: isolated in this study was named here as P. cinnamomi strain CPO-PCU (GenBank accession number JQ266267). It was identified by the BLASTn algorithm and the distance.
Point 20: Line 151: “colony”
Response 20: was removed.
Point 21: Line 152: “morphology”
Response 21: was inserted.
Point 22: Line 163-164: parenthesis removed.
Response: was removed.
Point 23: Line 186 “Aspect”
Response: was inserted.
Point 24: Line 188 “described”
Response: was inserted.
Point 25: Line 202: superindex 106
Response: was corrected.
Point 26: Line 212: deleted “in the proportion”
Response: was deleted
Point 27: Line 213: “1 L”
Response: was inserted.
Point 28: Line 213: drymifolia,
Response: was corrected, was italicized in the full text.
Point 29: Line 240: “Phytophthora sp. in vitro”
Response: was corrected, was italicized in the full text.
Point 30: Line 249: ...”they represented”
Response: was inserted.
Point 31: Line 251: ...” drymifolia”
Response: was italicized.
Point 32: Line 255: ...” Hypomyces”
Response: was italicized.
Point 33: Line 261: ... were characterized by the...
Response: was inserted.
Point 34: Line 265: “subglobose”
Response: was corrected.
Point 35: Line 266: “T. asperellum”
Response: was italicized.
Point 36: Line 270: “colonies of”
Response: was corrected.
Point 37: Line 270: “already mentioned above”
Response: was corrected.
Point 38: Line 274: “from”
Response: was corrected.
Point 39: Line 283: in 3.4 Percentage of “growth” inhibition in vitro
Response: was corrected.
Point 40: Line 305: “and they”
Response: was corrected.
Point 40: Line 305: “and”
Response: was corrected.
Point 41: Line 311: ...”facing up”...
Response: was corrected.
Point 42: Line 312: ...”they were”....
Response: was corrected
Point 43: Line 323: In Figure 5: “Hyphal interactions”
Response: was corrected.
Point 44: Line 329. “Trichoderma”
Response: was corrected, was italicized.
Point 45: Line T. koningiopsis
Response: was corrected, was italicized.
Point 46: Line: 350: “leaded”
Response: was corrected.
Point 47: Line 349: “days”
Response: was corrected.
____
Was deleted (Line 354): “Regarding the severity scale”...
Point 48: Line 370: “in vitro”
Response: was corrected, italicized.
Point 49: Line 371: “against”
Response: was corrected.
Point 50: Line 373: ... “genus” Trichoderma..
Response: was corrected.
Point 51: Line 373: well known as green- spored fungi,
Response: was corrected.
Point 52: Line 373: have been “reported” widely
Response: was corrected.
Point 53: Line 375: exhibit different
Response: was corrected. Inserted.
Point 54: Line 379: “can be”
Response: was corrected. Inserted.
Point 55: Line 393: “inside”
Response: was corrected. Inserted
Point 56: Line 418: in vitro
Response: was italicized.
Point 57: Line 418: “inter-action”
Response: was corrected.
Point 58: Line 423: “in root endophytes or”
Response: was corrected.
Point 59: Line 423: “in root endophytes or”
Response: was corrected.
Point 60: Line 428: the compounds isolated from the endophytic fungus T. koningiopsis (10S-7-isopropyl-4,10-dimethylbicyclo)
Response: was corrected.
Point 61: Line 439. In vitro.
Response: was italicized.
Point 62: Line 440: “causal”
Response: was corrected.
Point 63: Line 441: References
Response: was corrected.
Supplementary file titles:
Point 64: Table 1. Supplementary: Trichoderma
Response: was italicized.
Point 64: Figure 1. Supplementary: Seedling inoculated
Response: was italicized.